# Efficacy of a smartphone app to improve mental health among emergency service workers: A randomised controlled trial

Mark Deady[1]*, Mikayla Gregory[1], Quincy J. J. Wong[2], Denise Meuldijk[1], Daniel A. J. Collins[1], Lasse B. Sander[3], Richard Bryant[4], Samuel B. Harvey[1]

1 Black Dog Institute, Faculty of Medicine & Health, University of New South Wales, Sydney, New South Wales, Australia, 2 School of Psychology, Western Sydney University, Sydney, New South Wales, Australia, 3 Medical Psychology and Medical Sociology, Faculty of Medicine, University of Freiburg, Freiburg im Breisgau, Germany, 4 School of Psychology, University of New South Wales, Sydney, New South Wales, Australia

* m.deady@unsw.edu.au

## Abstract

### Background

Emergency service workers (ESWs) are routinely exposed to highly stressful and potentially traumatic events leading to high rates of psychological distress. Early intervention is vital to prevent chronic impairment and/or psychiatric disorders, with digital health innovations (e.g., smartphone apps) offering a potential means of scaling such intervention while overcoming barriers to help-seeking in this population. This study aims to evaluate the efficacy of an app designed to reduce psychological distress and related outcomes in ESWs.

### Methods

We conducted a randomised controlled trial with ESWs experiencing psychological distress (Kessler Psychological Distress Scale; K10) score >15. Participants were assigned to the intervention group, a full version of the *Build Back Better* app (including mindfulness, behavioural activation, trauma-focused cognitive therapy skills, activity and mood monitoring, and healthy coping strategies) or a mood/activity tracking-only version of the same app. Assessment occurred via online self-report questionnaires at baseline (T0) and at 1- and 3-month post-baseline (T1, T2) time-points. The primary outcome was the K10 score at T2. Linear mixed model analyses were conducted based on the intention-to-treat principle.

### Results

N = 880 ESWs were randomized to the full (n = 440) or tracking-only (n = 440) condition. There was no Time x Condition effect for K10 scores, with both conditions

**Data availability statement:** A minimal, deidentified dataset is available via UNSWorks public repository (https://doi.org/10.26190/unsworks/31622).

**Funding:** This research was funded by the Department of Health and Aged Care for the National Emergency Worker Support Service (Grant: 4-IHO1Z4B). SBH (grant number 1178666) and RB (grant no. 1173921) are supported by National Health and Medical Research Council (NHMRC) investigator grants. The funding sources had no role in the study design, data collection and analysis, decision to publish, or preparation of the manuscript.

**Competing interests:** MD, DM, DAJC, and SBH developed the Build Back Better app. They receive no financial benefit from this program. This does not alter our adherence to PLOS ONE policies on sharing data and materials.

showing similar improvements from baseline to 1-month (all $ps < .001$), and 3-month follow-up (all $ps < .001$). Exploratory analyses of moderators (engagement; baseline severity) found generally no significant differences in K10 score decreases across different levels of engagement in the full-app condition, however, app use was markedly low overall. At higher baseline psychological distress levels, there was a significant decrease in posttraumatic stress disorder symptoms from baseline to 1-month for the full-app group but not the tracking-only group ($p = .002$, d = −1.43).

## Conclusion

Despite consistent improvement across both app conditions, the minimal between group differences found here highlight the difficulties in developing effective, scalable resources for ESWs and the limitations of unguided digital programs more broadly.

## Trial registration

Australian New Zealand Clinical Trials Registry ACTRN12622001324707

## Introduction

Emergency service workers (ESWs), including police, ambulance personnel, firefighters and state emergency workers, have increased rates of mental health concerns compared to general population [1–3]. Prevalence rates of both depression and anxiety are as high as 15% among police, ambulance and firefighters [4,5], while one in ten ESWs report symptoms consistent with posttraumatic stress disorder (PTSD) [6].

In addition to the impact of potentially traumatic events, ESWs can experience other known psychosocial risk factors as part of their roles including high job demands, occupational violence, interpersonal issues, and shift work [3,7,8]. These hazards are compounded by poor help-seeking behaviours [9–11]. Stigma [9–11] and concerns that disclosure of mental health issues could influence career outcomes [12,13] present major reasons for this lack of help-seeking. Furthermore, there is an identified need for low-intensity interventions targeting ESWs due to the high-risk nature of their roles and the exposure to disaster and trauma, especially where these workers do not qualify for a formal mental health diagnosis [14].

Digital mental health interventions are a tool that has been developed to help overcome typical accessibility barriers in accessing psychological treatment for mental health issues [15]. These interventions deliver therapeutic evidence-based information via activities and interactive and engaging stimuli to help the user develop skills to assist them in managing their mental health [16,17]. Offering such programs via digital platforms has shown demonstrated effectiveness for prevention and management of mental ill-health symptoms [18–20]. However, less is known regarding app-based interventions [21]. Furthermore, evidence suggests that ESWs express interest and openness to digital health interventions [22] and apps specifically [23]. This study found that 54% would be interested in a mental health app for ESWs and this number increased to two thirds when the app targeted physical health. With

respondents favouring a mental health approach that leveraged the concept of "mental fitness". Despite this there have been relatively few studies investigating the efficacy of digital interventions for ESWs, and little direction around what to offer these workers [24–26]. Moreover, there is a lack of guidance around how to address subclinical symptoms in this high-risk, trauma-exposed group in order to avoid chronic functional and clinical impairment that comes with ingrained, untreated conditions [27].

The smartphone app *Build Back Better* was developed and tailored specifically with and for ESWs to improve psychological health and reduce symptoms of psychological distress and related outcomes. *Build Back Better* app was adapted from an existing app, *HeadGear* [28]. *HeadGear* previously underwent a pilot study, RCT, and a naturalistic evaluation, which collectively indicated effectiveness in preventing and improving depressive symptoms in both the general population and in male-dominated industries [28,29]. In order to adapt the program to the needs of ESWs an iterative codesign process occurred with ESWs, mental health experts including psychiatrists and clinical psychologists, consulted alongside design, user experience, and IT teams. Workshops and multiple rounds of user testing and revision took place during this phase with 12 ESWs to tailor this product. This codesign was deemed essential to meet the unique needs of the population. In order to overcome known challenges of engagement ([30]), several specific techniques associated with increased engagement were employed. These included personalised content where users selected area of need, data visualisation (tracking), reminders/push notifications, educational information, self-monitoring, and goal-setting features [31].

A pilot study of the app was previously conducted using an earlier version of the app to examine the acceptability, uptake and preferences of ESWs [32]. Feedback from the pilot study was further used to inform program changes. This primarily focused on elements of the user interface and simplifying the user flow through differing app screens to reduce cognitive burden and simplify navigation. Home screen was enhanced and elements of language within activities was simplified.

The current study aimed to evaluate the efficacy of the *Build Back Better* app in reducing symptoms of psychological distress and related outcomes in the ESW population by conducting a large scale randomised controlled trial (RCT) with ESWs across Australia.

## Methods

### Registration and approval

This study received ethical approval from The University of New South Wales Human Research Ethics Committee (HC220566). Consent was informed and obtained in writing (digitally) from all participants. The trial was prospectively registered via the Australian New Zealand Clinical Trials Registry: ACTRN12622001324707. Reporting was in line with CONSORT [33] and CONSORT e-Health [34] guidelines. Fig 1 presents the CONSORT flow diagram.

### Study design

This study was a two-armed RCT with the intervention group receiving the "full version" of the *Build Back Better* app and a tracking-only version of the app (mood and activity monitoring only), with a 2:2 ratio block allocation (4 participants per block). Assessments occurred at baseline (T0), 4 weeks post baseline (T1) and 3-month follow-up (T2) and took place between 01/03/2023 and 30/11/2023.

### Procedure

ESWs were recruited via both organisational and social media advertisements, which displayed QR codes and links to the study landing page where participants were required to provide informed consent before proceeding further. Individuals who consented to participate were screened for eligibility (see below) and either progressed to a web page informing them of their ineligibility (with a referral to Black Dog Institute's National Emergency Worker Support Service website containing additional resources), or, if eligible, they continued to the online baseline assessment (completed in the same session).

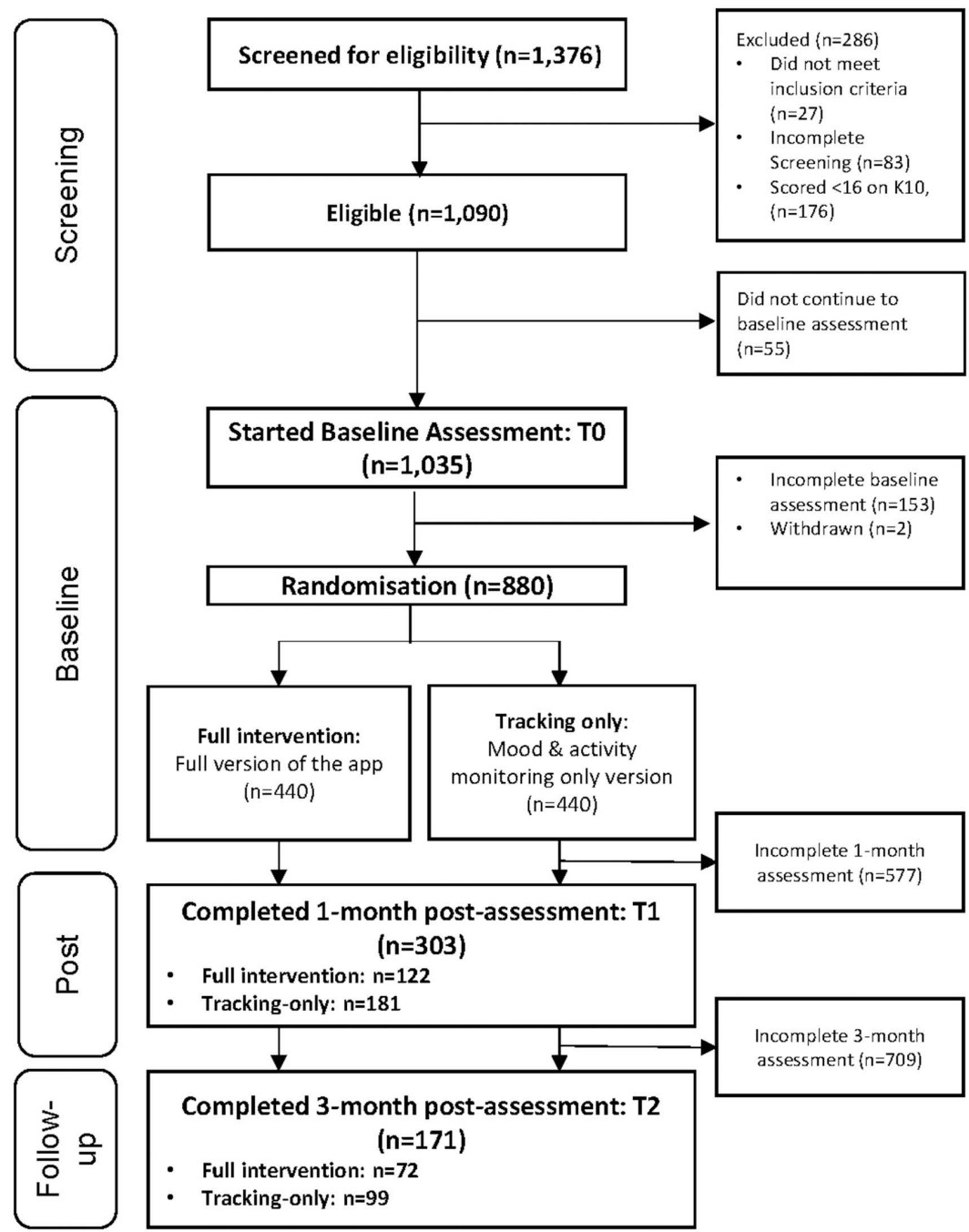

**Fig 1. Flow of participants through the trial.**

The baseline assessment included the Patient Health Questionnaire (PHQ-9) [35]; the trial safety protocol required the research team be notified if a participant scored 14 or higher on the PHQ-9, or indicated suicidal ideation on this measure, in which case the researchers followed up with participants and provided further support, including the opportunity to speak with a psychologist or mental health nurse to confirm they were not at heightened risk (29 participants were

subsequently called by the clinics and no participant was excluded following this call). Upon completion of all baseline measures (see Table 1), participants were randomised to either the full-intervention or tracking-only version of the app. Randomisation was performed using a computer-generated algorithm integrated into Black Dog Institute's digital trial management platform. All participants and investigators were masked from condition assignment until the completion of the final survey. Upon allocation, all study participants were given instructions on how to download the *Build Back Better* smartphone app including a login code that would take them to the appropriate app version (depending on their group allocation). Participants were asked to use the app consistently for at least 30 days. Participants received an email and text messages to complete follow-up surveys after 30 days and again at 3 months post-baseline, with up to two SMS and email reminders delivered at each timepoint if the survey was not completed.

## Participants

Eligibility criteria included being over the age of 18, currently or in the past 2 years having worked as an ESW, currently residing in Australia, owning a smartphone and having good English comprehension. The eligibility criteria also required participants to have scored 16 or higher on the Kessler Psychological Distress Scale (K10), indicating that they were experiencing at least moderate levels of psychological distress.

## Intervention arms

**Full intervention app.** The *Build Back Better* app (Fig 2) was developed as outlined above. Clinical aspects of original *HeadGear* [28] content including mindfulness, behavioural activation, mood monitoring, and healthy coping were maintained, while adaptations were made to include trauma-focused content including cognitive therapeutic techniques. The rationale for the clinical components of *HeadGear* are outlined in prior publications [36]. The addition of trauma-focussed cognitive therapeutic techniques was guided by best practice in the area [37], providing psychoeducation around

**Table 1. Assessment schedule.**

|  | T0* | T1 | T2 | In-app |
|---|---|---|---|---|
| K10 | x | x | x |  |
| Demographics (including lifetime trauma exposure and prior help-seeking) | x |  |  |  |
| PHQ-9 | x | x | x |  |
| GAD-7 | x | x | x |  |
| WHO-5 | x | x | x |  |
| CSE-T | x | x | x |  |
| PTSD-8 | x | x | x |  |
| AQoL-4D | x | x | x |  |
| CD-RISC-10 | x | x | x |  |
| HPQ | x | x | x |  |
| AUDIT-C | x | x | x |  |
| User Engagement/Feedback |  | x |  |  |
| Objective app engagement measures |  |  |  | x |

* Screening and baseline occurred in same session (T0).

Note. K10 = Kessler Psychological Distress Scale; PHQ-9 = Patient Health Questionnaire-9; GAD-7 = Generalized Anxiety Disorder-7; WHO-5 = World Health Organization Wellbeing Index; CSE-T = Trauma Coping Self-Efficacy Scale; PTSD-8 = Post-Traumatic Stress Disorder 8-item; AQoL-4D = Assessment of Quality of Life 4-dimension version; CD-RISC-10 = 10-item Connor-Davidson Resilience Scale; HPQ = Health and Work Performance Questionnaire; AUDIT-C = Alcohol Use Disorders Identification Test-Concise.

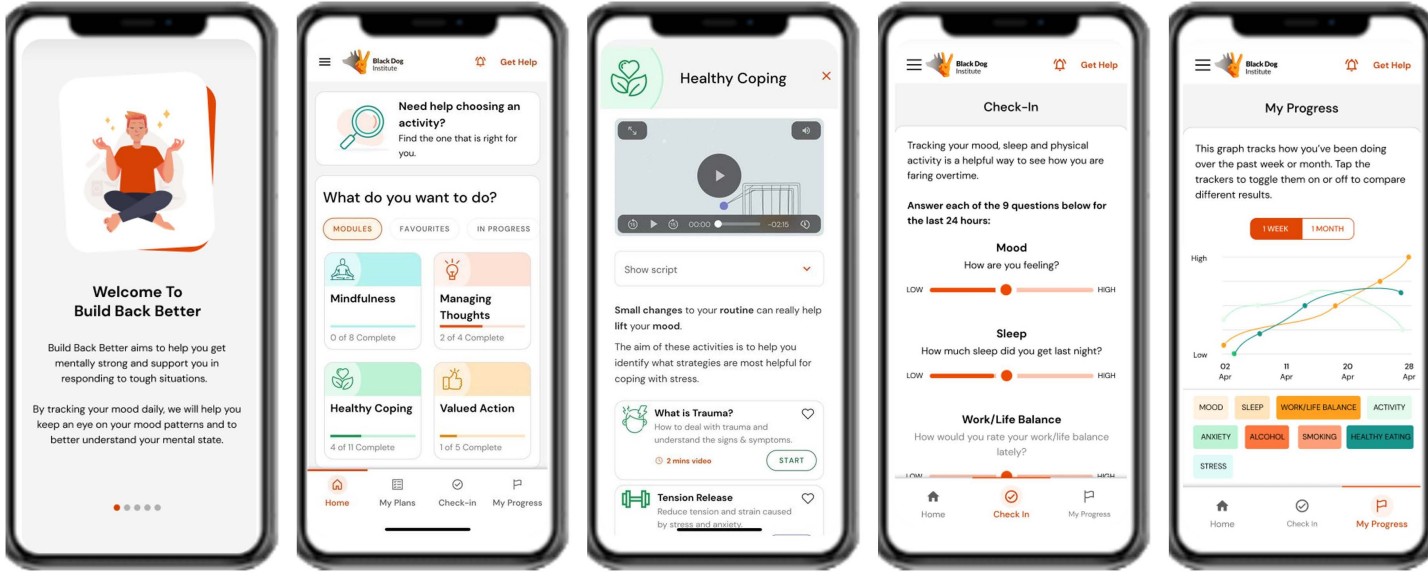

**Fig 2. Screenshots from *Build Back Better* app (intervention version).**

the nature of trauma responses and action around thought monitoring, labelling, and challenging. The format of the program was significantly altered to be less structured and allow for greater flexibility, based on ESW feedback. The user interface was also redesigned to meet the needs and preference of the ESWs. For complete list of activities see S1 File.

The *Build Back Better* app includes four key modules: 1) mindfulness, 2) healthy coping, 3) managing thoughts and 4) valued actions. Each module contains a combination of skill-based activities designed to deliver evidence-based therapeutic techniques via psychoeducational videos, audio recordings, and interactive exercises. The *Build Back Better* app also has a mood and activity tracker (encompassing mood, sleep, alcohol intake, smoking, physical activity, work/life balance, healthy eating, anxiety and stress) where users can log relevant variables on a slider scale daily, and view progress and correlations over time. The app also provides links and phone numbers for a variety of mental health and workplace support services.

**Tracking-only version.** The tracking-only (active control) version of the *Build Back Better* app had the same look and feel as the intervention version, however, the therapeutic skills-based activity content was not included. The tracking-only participants only had access to the mood/activity tracking component, where participants were encouraged to log their mood and activities over the 30-day trial period. They also had access to the graphing functions that displayed their mood and activities over the period of use. This app version also contained the same information page as the full intervention version with the links and phone numbers for a variety of mental health and workplace support services.

## Measures

At baseline, participants were asked a series of questions pertaining to demographics, lifetime trauma exposure and their previous help-seeking behaviour. Several validated scales were administered at baseline and follow-up timepoints (see Table 1), and these are described in detail below.

The primary outcome, psychological distress, was measured using the Kessler Psychological Distress Scale (K10), a 10-item measure of psychological distress over the past four weeks [38]. The K10 has a high level of internal consistency (Cronbach's $\alpha = 0.92$) and has been validated in large community samples with psychometric properties remaining consistent across a range of sociodemographic groups [39]. The K10 includes depression and

anxiety-based questions and answers are scored on a scale of 1–5 [38]. A total score is calculated by summing the scores across the 10 questions, with a minimum total score of 10 (no psychological distress) and a maximum of 50 (indicating severe distress) [38].

The Patient Health Questionnaire (PHQ-9) was used to measure depressive symptom severity [40]. This 9-item self-report scale has excellent internal consistency (Cronbach's $\alpha = 0.86$–$0.89$) and good criterion and construct validity as a measure of depression severity within large primary care samples [40], along with strong convergent validity with other well-established measures of depression ($r = 0.73$) when used in the general population [41]. The PHQ-9 asks participants to rate how often over the past two weeks they have been bothered by each of the 9 items on a 4-point Likert scale from 0 ("not at all") to 3 ("nearly every day") [35]. Total scores range from 0–27 with higher scores indicating more severe presentations.

The Generalized Anxiety Disorder-7 inventory (GAD-7) was used to measure symptoms of anxiety [42]. The GAD-7 has excellent internal consistency (Cronbach's $\alpha = 0.92$), strong construct validity (with higher scores associated with increasing functional impairment), and good convergent validity with other established anxiety scales ($r = 0.72$; $r = 0.74$) [42]. The GAD-7 asks how often participants have been bothered by 7 items, each rated on a 4-point Likert scale from 0 ("not at all") to 3 ("nearly every day") [42]. Total scores range from 0–27. The cutoffs on this scale include 5 (mild), 10 (moderate) and 15 (severe), with scores over 10 suggestive of an anxiety disorder being present.

Wellbeing was measured using the World Health Organization Wellbeing Index (WHO-5) [43,44]. The WHO-5 has good construct validity and is recommended for use in controlled trials to measure subjective wellbeing over time [45,46]. This 5-item scale asks participants to indicate which is closest to how they have been feeling over the past two weeks on a 6-point Likert scale from 0 ("at no time") to 5 ("all of the time") [45,46]. Raw scores therefore range from 0–25 with total scores calculated by multiplying the raw score by 4 giving a total score out of 100. Scores of 50 or less indicate poor wellbeing and scores of 28 or less are indicative of potential depression [45,46].

The Post-Traumatic Stress Disorder-8 item (PTSD-8) [47] is a validated short self-report screening tool for measuring probable ICD-11 defined PTSD [48,49]. The PTSD-8 has been validated in a range of trauma-exposed samples and shown to have good internal consistency (Cronbach's $\alpha = 0.83$–$0.85$) and a high level of concurrent validity with an established measure of trauma symptoms (Trauma Symptom Checklist; $r = 0.58$–$0.78$) [47]. After establishing Criterion A, the following 8 items represent symptoms across three subscales; intrusions (4 items), avoidance (2 items) and hypervigilance (2 items) [47]. Items are rated on a 4-point scale from 1 ("not at all") to 4 ("most of the time"). Probable PTSD diagnosis is met if at least one item in each of the three subscales has a score of 3 or 4 (i.e., participant reports that the symptom is experienced "sometimes" or "most of the time") [47,49].

Participants were also asked to rate their capacity for trauma-related coping using the Trauma Coping Self-Efficacy Scale (CSE-T) [50]. The CSE-T has sound psychometric properties across different trauma-exposed groups, with good test-retest reliability, internal reliability, and convergent and discriminant validity [50]. This 9-item scale asks participants to rate their perceived capacity for managing possible posttraumatic experiences (i.e., ability to deal with emotions since the traumatic event or capacity for controlling thoughts about the traumatic experience), on a 7-point scale that ranges from 1 ("not capable at all") to 7 ("totally capable"). Total scores are calculated by summing each of the items and range from 9 to 63 [50].

Work performance was measured using items from the Health and Work Performance Questionnaire (HPQ) [51]. Where participants reported being employed, they were asked to rate their perceived overall job performance over the previous four-week period on a scale of 1 ("worst performance") to 10 ("top performance"). They were also asked about absenteeism over the last month, the length of any absences and whether any absences were due to their mental health [51]. A score for effective workdays was calculated by subtracting the number of sick days taken over the past 4 weeks from 28 and multiplying by self-reported work performance rating (converted to a score from 0.1 to 1.0). This is consistent with previous studies [29,52].

Resilience was assessed using the 10-item Connor-Davidson Resilience Scale (CD-RISC-10), which has been shown to have excellent internal consistency (Cronbach's $\alpha = 0.89$) and test-retest reliability, as well as good convergent and discriminant validity [53,54]. The measure consists of 10 items asking participants to rate how they have felt over the past month (e.g., "under pressure, I stay focused and think clearly", "able to adapt when changes occur", or "I am able to handle unpleasant or painful feelings like sadness, fear and anger") on a 5-point scale from 0 ("not true at all") to 4 ("true nearly all of the time"). Total scores range from 0–40, with higher scores indicating greater resilience [53].

The Alcohol Use Disorders Identification Test-Concise (AUDIT-C) was used to screen participants for hazardous or problematic alcohol use [55]. This tool has excellent internal consistency (Cronbach's $\alpha = 0.94$), predictive validity, and concurrent validity with a lengthier 10-item measure ($r = 0.97$) [56]. The AUDIT-C is a 3-item scale asking participants how often they have a drink that contains alcohol, how many alcoholic drinks they consume on a typical day and how often they consume 6 alcoholic drinks or more on a single occasion [55]. Each of the 3 questions are scored on a 5-point scale ranging from 0–4 with the highest possible score of 12. Females with a score of 3 or more and males with a score of 4 or more is considered positive identification of hazardous drinking behaviours [55,57].

Participant quality of life was measured using the Assessment of Quality of Life (4 dimensions; AQoL-4D) [58]. This is a 12-item instrument with good internal consistency (Cronbach's $\alpha = 0.94$) and construct validity [58]. Questions are evenly grouped into 4 broad dimensions: independent living (e.g., "do you need help looking after yourself?"), mental health (i.e., questions regarding sleep, pain and anxiety/depression/worrying), relationships (with family, friends, etc.) and senses (i.e., vision and hearing related questions) [58,59]. Each item is rated on a 4-point scale ranging from 1 (lowest) to 4 (highest). Total unweighted scores are calculated for each of the 4 dimensions by subtracting 3 from the total score of that dimension and then dividing this number by four before multiplying it by 100 [58,59].

At the 1-month follow-up timepoint, participants were furthermore asked to complete 12 user engagement and app-related feedback questions which involved a combination of 5-point Likert scales and multiple choice questions and free text response questions. Items were adapted from the Mobile Application Rating Scale [60], including ease of use, understanding of content, engagement and interest in the design and content, likelihood of recommending to others, and overall rating of the app. Further questions measured the subjective perception of improvement in mental fitness, and reasons for stopping app use. Participants also provided general feedback and suggestions via open-response questions. These measures have been used previously and is provided in S2 [52].

App usage data was also collected throughout the research trial period pertaining to number of times the app was opened, the length of time spent in the app, the number of activities both accessed and completed as well as the number of mood and activity sessions that were logged by participants.

## Sample size

As a population-based, unguided, indicated intervention, the effect size of the intervention was anticipated to be relatively small. Based on review of the available literature of comparable studies [61–64] a small between-group effect size (Hedges g = 0.29) was expected. With alpha was set at 0.05 and power (1-β) at 0.80, and assuming a large correlation between repeated measures (i.e., $r > .99$) to have enough power to cover scenarios where the correlation would be smaller, a minimum of 394 participants was required per study arm. Allowing for a dropout rate of 30% of study participants from baseline, an estimated sample size of 1,126 was required for randomization (i.e., 563 participants per arm). G*Power was used to calculate power.

## Data analysis

Analyses were undertaken on an intent-to-treat basis. Linear mixed models for repeated measures analyses with maximum likelihood estimation were used to examine primary and secondary outcomes across baseline, 1-month and 3-month follow-ups. Maximum likelihood estimation incorporates all available data, including participants with missing

follow-up data, under the missing-at-random assumption. In the analytical models, two Time variables (reflecting baseline to 1-month, and 1-month to 3-month periods), Condition (full-app vs tracking-only), and corresponding Time x Condition interactions were specified as fixed effects, and a random intercept for participants was also included. Models did not adjust for baseline scores as randomisation in this trial resulted in comparable groups of participants in trial arms at baseline (see Table 2). An identity covariance matrix was employed, and degrees of freedom were estimated using Satterthwaite's method. Effect sizes (Cohen's d) were calculated based on relevant modelled mean differences and SDs at relevant timepoints. In addition to primary and secondary analyses, post-hoc exploratory analyses examined the effect of medication and help-seeking on the primary outcome results. Analyses were also conducted to investigate the moderating effect of app engagement and baseline severity on outcomes. Engagement was categorised as non-engagers, minimal engagers, and engagers; this was defined by completion of 0, 1–10, and >10 activities respectively.

For statistical significance, alpha was set at ≤ 0.05. SPSS Statistics 27.0 was used to conduct the analyses.

## Results

### Sample characteristics

Eight hundred and eighty ESWs met eligibility criteria, completed their baseline surveys and were randomly allocated to the full-app intervention or tracking-only condition. Of the baseline sample, 303 (34.4%) completed all outcome measures at 1-month follow-up, and 171 (19.4%) at 3-month follow-up (Fig 1).

Table 2 presents the baseline characteristics of the full study sample. The participants (n = 880) included ESWs from Fire and Rescue (30%), Ambulance (and other emergency medicine workers) (28%), Police (17%), State Emergency Services (13%), Surf Life Saving (2%), and 'Other' (10%). The mean age was 44.98 years (SD = 13.19), and the majority identified as male (60.9%). Just over half had been in emergency service work for 10 + years (55.2%) and a majority of the sample had experienced a traumatic event (88.5%). The majority had sought help from a mental health professional or GP during their lifetime (82.4%), with 31.6% taking medication for a mental health issue at trial registration and 39.1% engaging in help-seeking in the four weeks before trial registration.

Across all outcome measures, Little's Missing Completely at Random (MCAR) test was not significant, $\chi^2(3564)$ = 2379.28, p = 1.00, indicating missing data were MCAR. However, attrition from assessment was greater in the full intervention condition than the tracking-only condition at 1-month follow-up (72.3% vs 58.9%; $\chi^2(1)$ = 17.52, p < .001) and at 3-month follow-up (83.6% vs 77.5%; $\chi^2(1)$ = 5.29, p = .027). As the full sample of 1,024 could not be met due to recruitment constraints, the adjusted power was 0.76. In addition, missingness was significantly related to variables in the dataset (e.g., GAD-7 item and AUDIT-C item scores; ps < .046). Taken together, these results suggest missing data were more plausibly missing at random (MAR), and this aligns with the missing-at-random assumption of the linear mixed models approach for analyses.

Scores on all outcome measures approximated normality (absolute skewness < 3 and absolute kurtosis < 10; Kline, 2016) with the exception of the AQoL-4D independent living variable at the baseline and 1-month follow-up timepoints. AQoL-4D independent living variable raw scores and transformed scores that approximated normality were both analysed. Results from both analyses had the same pattern of significance and led to the same conclusions. Hence, the results from the analysis with the untransformed variable are reported.

### Primary outcome

Model-based estimates and Time x Condition interaction effects are presented in Table 3. There was no Time x Condition effect for scores on the K10, with both conditions showing similar improvements from baseline to 1-month (Bs ranged from −2.67 to −2.60, all |ts| > 6.10, all ps < .001, ds ranged from −0.43 to −0.42), and 3-month follow-ups (Bs ranged from −2.00 to −2.89, all |ts| > 4.35, all ps < .001, ds ranged from −0.46 to −0.32). For completeness, the difference in K10 scores between the two conditions was not significant at the 1-month and 3-month follow-ups (with respective Bs = 0.37 and

**Table 2. Participant characteristics at baseline.**

| | Full sample (N = 880) | Tracking-only (n = 440) | Full intervention (n = 440) |
|---|---|---|---|
| Age in years, M (SD, range) | 44.98 (13.19, 19–81) | 44.38 (13.23, 19–81) | 45.58 (13.14, 20–77) |
| Gender identity, n (%)[a] | | | |
| Female | 325 (36.9) | 165 (37.5) | 160 (36.4) |
| Male | 536 (60.9) | 269 (61.1) | 267 (60.7) |
| Non-binary or other | 16 (1.8) | 6 (1.4) | 10 (2.2) |
| Education, n (%) | | | |
| Year 12 (equivalent) or less | 135 (15.3) | 69 (15.7) | 66 (15.0) |
| Trade or other certificate or diploma | 367 (41.7) | 185 (42.0) | 182 (41.4) |
| University Degree | 350 (39.8) | 174 (39.5) | 176 (40.0) |
| Other | 28 (3.2) | 12 (2.7) | 16 (3.6) |
| Live in metropolitan area, n (%) | 285 (32.4) | 153 (34.8) | 132 (30.0) |
| Emergency service worker status, n (%)[b] | | | |
| Current paid | 482 (54.8) | 246 (55.9) | 236 (53.6) |
| Current volunteer | 320 (36.4) | 163 (37.0) | 157 (35.7) |
| Retired | 74 (8.4) | 29 (6.6) | 45 (10.2) |
| Service length, n (%)[c] | | | |
| <1 year | 18 (2.0) | 11 (2.5) | 7 (1.6) |
| 1–5 years | 223 (25.3) | 105 (23.9) | 118 (26.8) |
| 6–10 years | 152 (17.3) | 88 (20.0) | 64 (14.5) |
| 10 + years | 486 (55.2) | 236 (53.6) | 250 (56.8) |
| Sought professional/clinical help in lifetime (mental health professional, GP), n (%) | 725 (82.4) | 369 (83.9) | 356 (80.9) |
| Engaged in help-seeking in last four weeks, n (%) | 344 (39.1) | 170 (38.6) | 174 (39.5) |
| Currently taking medication for a mental health issue, n (%) | 278 (31.6) | 140 (31.8) | 138 (31.4) |
| Experienced traumatic event, n (%) | 779 (88.5) | 388 (88.2) | 391 (88.9) |
| **Primary outcome** | | | |
| K10, M (SD) | 25.42 (6.25) | 25.27 (6.26) | 25.57 (6.25) |
| **Secondary outcomes** | | | |
| PHQ-9, M (SD) | 9.79 (5.49) | 9.79 (5.54) | 9.79 (5.45) |
| GAD-7, M (SD) | 7.47 (4.57) | 7.54 (4.61) | 7.41 (4.53) |
| PTSD-8, M (SD)[d] | 16.72 (5.92) | 17.12 (6.31) | 16.40 (5.60) |
| CD-RISC-10, M (SD) | 26.67 (6.95) | 26.41 (6.89) | 26.92 (7.00) |
| CSE-T, M (SD) | 42.70 (11.10) | 42.11 (11.05) | 43.28 (11.13) |
| WHO-5, M (SD) | 38.53 (20.45) | 38.93 (20.45) | 38.13 (20.47) |
| AUDIT-C, M (SD) | 3.74 (2.94) | 3.83 (3.02) | 3.64 (2.85) |
| AQoL-4D, M (SD) | 78.35 (10.79) | 78.35 (10.90) | 78.35 (10.69) |
| Independent living | 95.42 (10.81) | 94.90 (11.44) | 95.93 (10.12) |
| Relationships | 69.90 (19.66) | 70.08 (19.03) | 69.72 (20.29) |
| Physical senses | 87.50 (12.22) | 87.22 (12.90) | 87.78 (11.49) |
| Mental health | 60.58 (17.62) | 61.19 (17.12) | 59.97 (18.10) |
| HPQ, M (SD)[e] | 18.15 (5.91) | 18.18 (6.12) | 18.12 (5.69) |

[a]Three participants in the full intervention condition indicated 'prefer not to say'.

[b]Two participants in the tracking-only app condition and two participants in the full intervention condition indicated 'prefer not to say'.

[c]One participant in the full intervention condition indicated 'prefer not to say'.

*(Continued)*

**Table 2.** (Continued)

[d]Administered only to participants who indicated that they experienced, witnessed or were confronted with a stressful experience or traumatic event in the last month (n = 299 in full sample: n = 133 in tracking-only condition; n = 166 in full intervention condition).

[e]Composite measure of effective workdays was calculated by multiplying work performance score for days worked during the previous 28 days by the number of days present at work over the same period.

Note. K10 = Kessler Psychological Distress Scale; PHQ-9 = Patient Health Questionnaire-9; GAD-7 = Generalized Anxiety Disorder-7; PTSD-8 = Post-Traumatic Stress Disorder 8-item; CD-RISC-10 = 10-item Connor-Davidson Resilience Scale; CSE-T = Trauma Coping Self-Efficacy Scale; WHO-5 = World Health Organization Wellbeing Index; AUDIT-C = Alcohol Use Disorders Identification Test-Concise; AQoL-4D = Assessment of Quality of Life 4-dimension version; HPQ = Health and Work Performance Questionnaire.

**Table 3. Model-based estimates and standard error for primary and secondary outcomes at each timepoint by condition.**

| | Full intervention | | | Tracking-only | | | T1 assessment Time x Condition | | | T2 assessment Time x Condition | | |
|---|---|---|---|---|---|---|---|---|---|---|---|---|
| | T0 (n = 440) | T1 (n = 122) | T2 (n = 72) | T0 (n = 440) | T1 (n = 181) | T2 (n = 99) | statistic | df | p | statistic | df | p |
| K10, M (SE) | 25.57 (0.30) | 22.97 (0.47) | 22.68 (0.57) | 25.28 (0.30) | 22.60 (0.40) | 23.28 (0.50) | t = 0.13 | 603.82 | .898 | t = −1.26 | 599.27 | .207 |
| PHQ-9, M (SE) | 9.79 (0.26) | 8.69 (0.40) | 8.40 (0.49) | 9.79 (0.26) | 8.91 (0.35) | 8.70 (0.43) | t = −0.46 | 586.92 | .648 | t = −0.49 | 582.29 | .628 |
| GAD-7, M (SE) | 7.40 (0.22) | 6.28 (0.35) | 6.27 (0.43) | 7.54 (0.22) | 6.77 (0.30) | 6.63 (0.37) | t = −0.82 | 570.83 | .414 | t = −0.39 | 567.83 | .695 |
| PTSD-8, M (SD)[a] | 16.39 (0.45) | 13.59 (0.83) | 13.66 (1.04) | 16.98 (0.49) | 14.61 (0.73) | 15.26 (0.96) | t = −0.40 | 130.80 | .687 | t = −0.72 | 158.55 | .476 |
| CD-RISC-10, M (SE) | 26.92 (0.34) | 26.89 (0.50) | 26.92 (0.60) | 26.41 (0.34) | 26.88 (0.44) | 27.19 (0.55) | t = −0.88 | 497.13 | .378 | t = −0.78 | 490.95 | .292 |
| CSE-T, M (SE) | 43.28 (0.54) | 44.06 (0.85) | 45.43 (1.03) | 42.11 (0.54) | 44.65 (0.73) | 44.77 (0.91) | t = −1.75 | 537.28 | .081 | t = −0.39 | 531.20 | .694 |
| WHO-5, M (SE) | 38.13 (1.00) | 40.72 (1.66) | 37.43 (2.05) | 38.93 (1.00) | 41.41 (1.41) | 45.01 (1.80) | t = 0.05 | 587.97 | .960 | t = −2.57 | 584.11 | .010 |
| AUDIT-C, M (SE) | 3.64 (0.14) | 3.26 (0.17) | 3.34 (0.18) | 3.83 (0.14) | 3.57 (0.15) | 3.42 (0.17) | t = −0.85 | 426.95 | .397 | t = 0.60 | 424.67 | .550 |
| AQoL-4D, M (SE) | 78.35 (0.53) | 78.02 (0.76) | 78.92 (0.89) | 78.35 (0.53) | 78.58 (0.67) | 78.01 (0.82) | t = −0.69 | 463.22 | .488 | t = 0.86 | 457.74 | .390 |
| AQoL-4D IL, M (SE) | 95.93 (0.53) | 95.05 (0.80) | 94.43 (0.96) | 94.90 (0.53) | 94.98 (0.70) | 95.13 (0.87) | t = −1.04 | 480.70 | .297 | t = −1.47 | 474.09 | .143 |
| AQoL-4D Rel, M (SE) | 69.72 (0.95) | 69.90 (1.46) | 71.21 (1.74) | 70.08 (0.95) | 70.40 (1.27) | 68.29 (1.59) | t = −0.09 | 480.40 | .929 | t = 1.52 | 473.75 | .129 |
| AQoL-4D PS, M (SE) | 87.78 (0.59) | 86.35 (0.99) | 89.22 (1.21) | 87.22 (0.59) | 86.94 (0.85) | 85.43 (1.10) | t = −0.93 | 532.69 | .351 | t = 2.06 | 525.26 | .040 |
| AQoL-4D MH, M (SE) | 59.97 (0.86) | 60.71 (1.36) | 61.06 (1.65) | 61.19 (0.86) | 61.95 (1.18) | 62.92 (1.50) | t = −0.02 | 498.46 | .986 | t = −0.31 | 491.29 | .755 |
| HPQ, M (SE) | 18.10 (0.33) | 17.75 (0.62) | 18.05 (0.79) | 18.18 (0.33) | 17.37 (0.50) | 18.16 (0.71) | t = 0.58 | 428.64 | .563 | t = −0.02 | 445.54 | .983 |

[a]Administered only to participants who indicated that they experienced, witnessed or were confronted with a stressful experience or traumatic event in the last month (tracking-only condition: n = 133 at baseline, n = 43 at 1-month follow-up, n = 22 at 3-month follow-up; full intervention condition: n = 166 at baseline, n = 31 at 1-month follow-up, n = 22 at 3-month follow-up).

Note. Time x Condition interactions reflect the period from baseline to the specified timepoint.

Note. K10 = Kessler Psychological Distress Scale; PHQ-9 = Patient Health Questionnaire-9; GAD-7 = Generalized Anxiety Disorder-7; CD-RISC-10 = 10-item Connor-Davidson Resilience Scale; CSE-T = Trauma Coping Self-Efficacy Scale; WHO-5 = World Health Organization Wellbeing Index; AUDIT-C = Alcohol Use Disorders Identification Test-Concise; AQoL-4D = Assessment of Quality of Life 4-dimension version; IL = Independent Living; Rel = Relationships; PS = Physical Senses; MH = Mental Health; HPQ = Health and Work Performance Questionnaire.

−0.60, both |ts| < 0.79, both ps > .428, d = 0.06 favouring tracking-only condition at 1-month follow-up and d = −0.08 favouring full intervention condition at 3-months follow-up).

### Secondary outcomes

Only a time effect was present for depression and anxiety symptoms (PHQ-9 and GAD-7), which significantly improved from baseline to the 1- and 3-month follow-ups in both conditions (Bs ranged from −1.39 to −0.77, all |ts| > 2.59, all ps < .010, ds ranged from −0.25 to −0.16). Alcohol consumption also showed a time effect, decreasing significantly from baseline to both follow-up timepoints in both conditions (Bs ranged from −0.41 to −0.26, all |ts| > 2.34, all ps < .020, ds ranged from −0.14 to −0.09).

PTSD symptoms showed no between group effects, generally decreasing from baseline to the 1- and 3-month follow-ups in both conditions (Bs ranged from −2.81 to −2.37, all |ts| > 2.58, all ps < .011, ds ranged from −0.47 to −0.40), although decrease from baseline to 3-month follow-up for tracking-only group was not significant (B = −1.72, 95% CI [−3.59, 0.14], t(120.44) = −1.83, p = 0.070, d = −0.29). Trauma coping self-efficacy generally improved from baseline to the 1-month and 3-month follow-ups in both conditions (Bs ranged from 2.14 to 2.65, all |ts| > 2.23, all ps < .026, ds ranged from 0.19 to 0.24), although increase from baseline to 1-month follow-up for the full intervention group was not significant (B = 0.78, 95% CI [−0.74, 2.30], t(545.23) = 1.01, p = 0.312, d = 0.07).

There was no differential Time x Condition effect evident in wellbeing from baseline to 1-month follow-up and no significant change in either condition (Bs ranged from 2.49 to 2.59, both |ts| < 1.88, all ps > .061, ds ranged from 0.12 to 0.13) for this period. However, from baseline to 3-month follow-up, wellbeing significantly increased in the tracking-only arm (B = 6.08, 95% CI [2.68, 9.48], t(571.84) = 3.51, p < .001, d = 0.30) but not the full intervention arm (B = −0.70, 95% CI [−4.61, 3.21], t(593.58) = −0.35, p = .726, d = −0.03), resulting in significantly higher wellbeing at 3-month follow-up favouring tracking-only (B = −7.58, 95% CI [−12.92, −2.23], t(936.67) = −2.78, p = .006, d = −0.32).

There were no time effects for work performance (Bs ranged from −0.80 to −0.02, all |ts| < 1.62, all ps > .106, ds ranged from −0.14 to 0.00), resilience (Bs ranged from −0.04 to 0.78, all |ts| < 1.59, all ps > .112, ds ranged from −0.01 to 0.11) or overall health-related quality of life (Bs ranged from −0.34 to 0.57, all |ts| < 0.73, all ps > .467, ds ranged from −0.03 to 0.05). There was also no change over time in the quality of life dimensions of independent living (Bs ranged from −1.50 to 0.23, all |ts| < 1.70, all ps > .089, ds ranged from −0.14 to 0.02), relationships (Bs ranged from −1.78 to 1.49, all |ts| < 1.24, all ps > .215, ds ranged from −0.09 to 0.08), and mental health (Bs ranged from 0.74 to 1.73, all |ts| < 1.25, all ps > .211, ds ranged from 0.04 to 0.10). There was no change in quality of life related to physical senses in either condition from baseline to 1-month follow-up (Bs ranged from −1.43 to −0.29, both |ts| < 1.52, all ps > .128, ds ranged from −0.12 to −0.02), and no differential Time x Condition effect evident for this period. However, from baseline to 3-month follow-up there was a significant differential Time x Condition effect (see Table 2). Neither the decrease in quality of life related to physical senses for the tracking-only condition (B = −1.79, 95% CI [−3.84, 0.27], t(516.98) = −1.71, p = .088, d = −0.15) nor the increase for the full intervention condition (B = 1.44, 95% CI [−0.85, 3.73], t(532.03) = 1.24, p = .217, d = 0.12) reached statistical significance. The full intervention group did, however, have significantly better mean scores on this dimension relative to the tracking-only group at 3-month follow-up (B = 3.78, 95% CI [0.58, 6.99], t(849.01) = 2.32, p = .021, d = 0.29).

### Post-hoc exploratory analyses

**Effect of medication and help-seeking on primary outcome results.** The primary outcome analysis was repeated but with Time x Condition x Medication interactions (one for each time period) and relevant lower order two-way interactions added as fixed effects into the model. This model tested whether the Time x Condition interactions were moderated by medication use for mental health at the time of trial registration. Results indicated the Time x Condition interactions in the primary outcome analyses were not moderated by medication use (for both periods baseline to 1-month follow-up and baseline to 3-month follow-up: |ts| < 0.93, ps > .352).

A similar model tested whether Time x Condition interactions in the primary outcome analyses were moderated by help-seeking in the four weeks prior to trial registration. Help-seeking significantly moderated the Time x Condition interaction for baseline to 1-month follow-up (B = −3.30, 95% CI [−5.59, −1.02], t(577.82) = −2.84, p = .005), but not for baseline to 3-month follow-up (B = −2.83, 95% CI [−5.73, 0.07], t(574.89) = −1.92, p = .056). The significant moderation for baseline to 1-month follow-up reflected a non-significant Time x Condition interaction when there was help-seeking prior to the trial (B = −1.64, 95% CI [−3.35, 0.06], t(251.63) = −1.89, p = .060; both tracking-only and full intervention conditions showed a significant decrease in K10 scores, with respective Bs = −2.26 and −3.90, both |ts| > 3.92, both ps < .001, ds = −0.36 and −0.62), and a significant Time x Condition interaction when there was no help-seeking prior to the trial (B = 1.60, 95% CI [0.06, 3.14], t(328.16) = 2.05, p = .041). With no prior help-seeking, the full intervention condition exhibited a significant decrease in K10 scores from baseline to 1-month follow-up (B = −1.50, 95% CI [−2.71, −0.29], t(336.33) = −2.44, p = .015, d = −0.24), but the tracking-only condition exhibited an even larger significant decrease in K10 scores (B = −3.11, 95% CI [−4.06, −2.16], t(315.33) = −6.44, p < .001, d = −0.50). However, the small sample size and missing data mean these findings should be interpreted with caution.

**App engagement analysis for the intervention condition – primary outcome.** Psychological distress significantly decreased from baseline to the 1-month follow-up similarly for non-engagers, minimal engagers, and engagers (Bs ranged from −3.01 to −1.87, all |ts| > 2.42, all ps < .016, ds ranged from −0.48 to −0.30), but there were no differential Time x App engagement effects evident for this period (see Table 4). Psychological distress also significantly decreased from baseline to the 3-month follow-up for non-engagers, minimal engagers, and engagers (Bs ranged from −4.68 to −1.84, all |ts| > 2.19, all ps < .030, ds ranged from −0.75 to −0.29), although one Time x App engagement interaction was detected as significant (see Table 4). Specifically, while the decrease in psychological distress for minimal engagers was significant (p = .030, d = −0.29), there was an even greater significant decrease for non-engagers (p < .001, d = −0.75). The resulting difference in psychological distress at 3-month follow-up between minimal engagers and non-engagers was not significant (B = 2.26, 95% CI [−0.44, 4.96], t(381.22) = 1.65, p = .100, d = 0.34). S7 provides group level characteristics at baseline for each group.

**Baseline psychological distress (K10) as moderator of Time x Condition effects for primary and secondary outcomes.** As shown in Table 5, baseline psychological distress was a significant moderator of the Time x Condition

**Table 4. Model-based estimates and standard error for primary outcomes at each timepoint in the intervention condition based on app engagement.**

| | Non-engager (n=229) (reference group) | | | Minimal engager (n=151) | | | T1 assessment | | | T2 assessment | | |
| | | | | | | | Time x App engagement | | | Time x App engagement | | |
| | T0 | T1 | T2 | T0 | T1 | T2 | statistic | df | p | statistic | df | p |
|---|---|---|---|---|---|---|---|---|---|---|---|---|
| K10, M (SE) | 25.79 (0.42) | 22.78 (0.78) | 21.10 (1.03) | 25.20 (0.51) | 23.33 (0.84) | 23.36 (0.90) | t=1.08 | 248.34 | .283 | t=2.18 | 245.90 | .030 |
| | Non-engager (n=229) (reference group) | | | Engager (n=60) | | | T1 assessment | | | T2 assessment | | |
| | | | | | | | Time x App engagement | | | Time x App engagement | | |
| | T0 | T1 | T2 | T0 | T1 | T2 | statistic | df | p | statistic | df | p |
| K10, M (SE) | 25.79 (0.42) | 22.78 (0.78) | 21.10 (1.03) | 25.68 (0.81) | 22.91 (0.85) | 23.07 (1.03) | t=0.24 | 230.61 | .811 | t=1.55 | 232.67 | .123 |
| | Minimal engager (n=151) | | | Engager (n=60) | | | T1 assessment | | | T2 assessment | | |
| | | | | | | | Time x App engagement | | | Time x App engagement | | |
| | T0 | T1 | T2 | T0 | T1 | T2 | statistic | df | p | statistic | df | p |
| K10, M (SE) | 25.20 (0.51) | 23.33 (0.84) | 23.36 (0.90) | 25.68 (0.81) | 22.91 (0.85) | 23.07 (1.03) | t=−0.88 | 226.86 | .380 | t=−0.63 | 227.87 | .529 |

Note. K10 = Kessler Psychological Distress Scale.

**Table 5. Statistics testing baseline psychological distress (K10) as moderator of Time x Condition effects for primary and secondary outcomes.**

| | T1 assessment | | | T2 assessment | | |
|---|---|---|---|---|---|---|
| | Time x Condition x Baseline psychological distress | | | Time x Condition x Baseline psychological distress | | |
| | *statistic* | *df* | *p* | *statistic* | *df* | *p* |
| K10, M (SE) | t=−0.51 | 1151.34 | .611 | t=−0.82 | 1201.28 | .415 |
| PHQ-9, M (SE) | t=−1.51 | 768.02 | .131 | t=−0.92 | 782.20 | .359 |
| GAD-7, M (SE) | t=−1.74 | 700.30 | .082 | t=−0.53 | 714.65 | .596 |
| PTSD-8, M (SD)ᵃ | t=−2.46 | 138.17 | .015 | t=1.15 | 181.81 | .253 |
| CD-RISC-10, M (SE) | t=0.55 | 515.11 | .580 | t=−1.07 | 510.94 | .284 |
| CSE-T, M (SE) | t=1.63 | 587.17 | .104 | t=−0.60 | 586.07 | .546 |
| WHO-5, M (SE) | t=−0.66 | 686.29 | .509 | t=−0.11 | 694.93 | .913 |
| AUDIT-C, M (SE) | t=0.50 | 426.72 | .616 | t=−0.79 | 425.06 | .428 |
| AQoL-4D, M (SE) | t=−1.54 | 489.32 | .124 | t=−0.13 | 484.40 | .898 |
| AQoL-4D IL, M (SE) | t=−0.82 | 485.77 | .415 | t=−1.21 | 480.87 | .228 |
| AQoL-4D Rel, M (SE) | t=−1.91 | 496.08 | .057 | t=0.63 | 491.11 | .531 |
| AQoL-4D PS, M (SE) | t=−1.01 | 534.90 | .311 | t=−1.06 | 530.47 | .289 |
| AQoL-4D MH, M (SE) | t=0.03 | 544.97 | .977 | t=0.71 | 540.93 | .479 |
| HPQ, M (SE) | t=0.64 | 456.43 | .524 | t=1.01 | 479.50 | .311 |

ᵃAdministered only to participants who indicated that they experienced, witnessed or were confronted with a stressful experience or traumatic event in the last month (tracking-only condition: n=133 at baseline, n=43 at 1-month follow-up, n=22 at 3-month follow-up; full intervention condition: n=166 at baseline, n=31 at 1-month follow-up, n=22 at 3-month follow-up).

Note. Time x Condition interactions reflect the period from baseline to the specified timepoint. All tested models contained relevant main effects, two-way interactions, and three-way interaction, but only the three-way interaction results are presented in the table.

Note. K10=Kessler Psychological Distress Scale; PHQ-9=Patient Health Questionnaire-9; GAD-7=Generalized Anxiety Disorder-7; PTSD-8=Post-Traumatic Stress Disorder 8-item; CD-RISC-10=10-item Connor-Davidson Resilience Scale; CSE-T=Trauma Coping Self-Efficacy Scale; WHO-5=World Health Organization Wellbeing Index; AUDIT-C=Alcohol Use Disorders Identification Test-Concise; AQoL-4D=Assessment of Quality of Life 4-dimension version; IL=Independent Living; Rel=Relationships; PS=Physical Senses; MH=Mental Health; HPQ=Health and Work Performance Questionnaire.

effect for PTSD-8 scores from baseline to 1-month follow-up only, and not for any other outcome at any other timepoints. To further examine this significant moderation effect, the Time x Condition effect for PTSD-8 scores from baseline to 1-month follow-up was evaluated at a baseline K10 score below the mean and a baseline K10 score above the mean. The mean baseline K10 score was 25.42 (SD=6.25), which fell in the moderate distress category. As such, the middle score of the mild distress category (baseline K10 score=22) and the middle score of the severe distress category (baseline K10 score=40) were used to further evaluate the Time x Condition effect for PTSD-8 scores.

At a baseline K10 score of 22 (below the mean), the Time x Condition effect for PTSD-8 scores from baseline to 1-month follow-up was not significant (B=−1.64, 95% CI [−4.00, 0.73], t(146) = −1.37, p=.173). However, at a baseline K10 score of 40 (above the mean), the Time x Condition effect for PTSD-8 scores from baseline to 1-month follow-up was significant (B=6.04, 95% CI [0.54, 11.50], t(129) = 2.17, p=.032). This significant Time x Condition effect reflected a significant decrease in PTSD-8 scores from baseline to 1-month follow-up for the full intervention arm (B=−5.15, 95% CI [−9.31, −0.98], t(141) = −2.44, p=.016, d=−0.86), but a non-significant change in PTSD-8 scores from baseline to 1-month follow-up for the tracking-only arm (B=0.89, 95% CI [−2.70, 4.48], t(114) = 0.49, p=.624, d=0.15). This resulted in significantly lower PTSD-8 scores at 1-month follow-up favouring the full intervention condition (B=−8.62, 95% CI [−14.00, −3.19], t(224) = −3.13, p=.002, d=−1.43). The small sample size and missing data mean these findings should be interpreted with caution.

## App use and feedback

App engagement was low overall, with the full intervention group, on average, logging into the app 4.66 (SD = 6.41) times over the instructed 30-day period, significantly less than the tracking-only group (9.53; SD = 10.00; p < .001). Engagement with the skills-based app content was also low, with the full intervention group accessing a mean of 4.78 (SD = 9.90) activities. Number of activity and mood monitoring sessions were also tracked, with tracking-only participants using the check-in function significantly more than full intervention group users (M = 19.93; SD = 4.64, compared to M = 6.88; SD = 8.67; p < .001). Overall, those completing the follow-up survey were more likely to have higher rates of use ($t_{128.29}$ = −7.40 p < .001).

All participants were asked at the 1-month follow-up to provide feedback on the app. In terms of the extent to which they felt the app had helped them to improve their mental fitness (from 1 "strongly disagree" to 5 "strongly agree"), full intervention participants rated the app as significantly more helpful on average (3.27/5; SD = 0.96) compared to tracking-only participants (2.56/5; SD = 0.87; p < .001). Almost all (91.35%) of the full intervention respondents indicated they would recommend the app to others, while only 66.89% of tracking-only respondents claimed they would do so (p < .001). Full intervention app users also rated the app significantly higher overall (4.07/5; SD = 0.93) compared to tracking-only app users (3.34/5; SD = 1.11; p < .001). Between-group comparisons for app user feedback items can be seen in S4 File.

Around half of respondents (129/254; 50.79%) reported stopping use of the app before completing the trial. A higher proportion of the full intervention group (61/103; 59.22%) ceased app use compared to the tracking-only group (68/151; 45.03%). The most common reason for stopping in the full intervention group was lack of time (23/61; 37.70%), whereas the most common reason in the tracking-only group was loss of interest (29/68; 42.65%),

When asked to nominate their favourite app feature, full intervention respondents most frequently noted ease of use/navigation, the wide variety of available content, and the daily tracker. Of the four key content areas, mindfulness was favoured the most.

In terms of least favourite app feature, full intervention respondents most frequently mentioned lack of reminders to use the app (this function was available but required setting changes to turn on), and lack of engaging or relevant content. A small minority reported technical issues (such as inability to access offline) and difficulty with navigation/ease of use.

## Discussion

This study reports on the evaluation of *Build Back Better*, a smartphone app designed to reduce symptoms of psychological distress and related outcomes in ESWs. The RCT was conducted with ESWs across Australia, from a range of emergency service roles. Overall, we found the full app to perform no better than a tracking-only version (in which users monitored their mood and behaviour). Users of both versions of the app improved on most outcomes over time.

The null finding provides valuable insight into the utility of such programs within this workforce. We found engagement to be low, particularly for the full intervention group. Despite this, those who responded to the 1-month follow-up questionnaire provided generally positive user feedback on the app, with strong overall scores and high rates of recommendation, and the app was viewed significantly more favourably by the full intervention group. Of course, there is a degree of bias within this finding as responders were more likely to engage more with the app itself. It is possible that despite satisfaction with the app itself, engagement was limited by other barriers known to impact ESWs, such as time constraints and restrictions due to organisational culture and shift work [65]. Regardless, evidence suggests positive user experience is only weakly correlated with sustained app engagement [66], and the favourable user feedback is insufficient to recommend wider rollout of this app to ESWs in its current form. Importantly, greater engagement was not associated with between-group differences. This is in contrast to consistent findings showing differential effects of similar content in more generalised workforces [29,52], especially at high levels of engagement. In these previous RCTs, rates of app use were more than double those seen here. However, these studies were focused on those with lower levels of symptoms. The present findings suggest that this form of brief, unguided digital intervention may not be adequate to see a marked difference in

change scores compared to a month-long tracking-only condition in this vulnerable group. However, low engagement overall makes it difficult to determine whether users received adequate 'dosage' within this study. This concept of adequacy of dosage remains an ongoing issue especially within self-guided programs. Indeed, there is some evidence to support this from a recent meta-analysis [67], at low levels of symptoms guidance does not increase effect sizes while more sensitive, guided, digital precision is more appropriate at higher symptom levels. Though the present sample was not severely unwell on the K10, there was considerable comorbid trauma symptomatology which may have introduced significant complexity into the clinical profile of this sample, requiring more intensive content or delivery. Similarly, the high rates of medication use and help-seeking indicate more pronounced impairment. Interestingly, help-seeking appeared to lead to significant decrease in K10 scores for the full intervention arm over the course of follow-up (although for those who were not seeking help, the tracking-only arm had improved short-term outcomes—see S3 File). Finally, in line with recommendations for codesign of ESW interventions [68], representatives from this population were involved in ongoing consultation throughout the app development process, and on the basis of iterative codesign *Build Back Better* moved from a sequential 30-day once-daily format to an unstructured, self-directed format which may have impacted the effectiveness of the app [69]. It is unclear exactly what drove this null finding especially where the clinical basis for the app was previously associated with effect [29]. However, two major distinctions were the baseline clinical profile of users and the freeform app format. Specifically, this format change may have led to poorer direction within the app. Although users in all these studies were directed to use the app consistently for 30 days, this instruction is likely to be considerably clearer where the app structure is aligned in this manner. Instead, it is more likely present study participants (especially as they were symptomatic at baseline) may have used the app flexibility (e.g., as 'needed'), which is unlikely to bring about sustained effect. Equally, while the control condition looked to mirror this flexibility, ecological momentary assessment may have resulted in improved engagement.

Alternate explanations for these findings include the potential effect of the tracking-only condition which may have led to real, comparable change in this arm of the study. Indeed, there is evidence to suggest mood monitoring is associated with symptom improvement particularly in the case of depression [70,71], with active controls of this nature associated with significantly smaller effect sizes compared to other forms of control [18]. This tool may have been further enhanced by the inclusion of other forms of monitoring (e.g., activity, sleep) which may have allowed users to recognise patterns in these factors and improve self-regulation. This adds to the ongoing debate around accuracy of evidence for psychological intervention in the context of appropriate control conditions, and, in fact, what such conditions may be [72]. Further comparisons of monitoring and specific self-guided program elements are required to better understand the utility of this form of control condition and what therapeutic impacts it may have.

Interestingly, we found more sustained change in PTSD symptoms in the full app condition, compared to the tracking-only arm. Furthermore, that at higher baseline levels of psychological distress, the full app condition was associated with a significant short-term decrease in PTSD symptoms, which was not present in the tracking-only version of the app. Suggesting that although the distress symptoms measured by the K10 may not have been sensitive to change, the specific PTSD symptoms experienced may have undergone some differential improvement when the full content was available. Again, the exploratory nature of these findings suggest further exploration is required to confirm these preliminary findings.

While there has been very limited research involving digital interventions with ESWs [68] and uptake and acceptability of interventions in this population are frequently reported as low [65], there is some evidence for Internet-based mindfulness in improving adaptive resilience in firefighters [73]. However, more broadly there some evidence to suggest that relaxation (sometimes conflated with elements of mindfulness) might not be beneficial for depression outcomes when delivered in digital programs [74], while behavioural activation is among the most effective components [74,75]. Nevertheless, although both mindfulness and behavioural activation formed core therapeutic content of the *Build Back Better* content, engagement with these activities may not have been sufficient to elicit effect. Relatedly, there is some emerging

evidence for the benefits of interventions that require body movement and group delivery within this high-risk workforce [76–78]. Indeed, it has been found that firefighters have a preference for behavioural components of therapy [23] while the use of human support [79] may also be more important among this occupational group than in others, especially due to the comparably lower rates of technology uptake and engagement in these groups [80,81].

This trial was limited by constrained recruitment, with randomised sample failing to meet the sample size estimate. Furthermore, this trial was limited by high rates of attrition, which may have impacted power to detect an effect. As noted above, app engagement was low. Previous work in this area has highlighted a number of elements that might enhance engagement with workplace digital interventions [82], many of which were incorporated in the app build, such as brief activities [83] and persuasive technology (e.g., reminders and self-monitoring) [84]. Nevertheless, the unstructured program design may have limited engagement in the intervention. A recent meta-analyses have demonstrated that despite the potential for persuasive design systems to positively influence behaviour change [85], the association between these principles and efficacy or engagement in mental health apps is a best mixed [86,87]. There were some untested elements which might have played into the findings including expectancy effects which are known to impact effectiveness of digital interventions. Similarly, as with any trial of this kind, there is the potential for a self-selection bias, and thus, the findings may not be representative of all workers. Finally, PTSD symptom findings were impacted by an error in questionnaire skip procedure meaning items were not asked of all users at follow-up (only those with past-month exposure).

This study found little support for a mindfulness and CBT digital intervention for ESWs experiencing psychological distress outperforming a tracking-only version of the app. These findings highlight the challenges of unguided interventions, especially in at-risk occupational groups. Despite previous findings and codesign work suggesting appeal within this group, engagement with the app was a major limitation. These low levels of use, combined with significant trial attrition, are likely to have impacted effectiveness. These challenges are not unique to ESWs but underscore the barriers to implementation of unguided interventions. Alternate avenues of intervention to address the mental health needs of these workers are required, particularly where traditional services are unavailable or where individuals do not meet clinical cutoff levels. These alternatives might include blended care or more guided modes of delivery (e.g., [88]). The use of adaptive and individually tailored approaches may also improve engagement and could be considered as a future direction of research.

## Supporting information

**S1 File. Intervention content.**
(DOCX)

**S2 File. Feedback items.**
(DOCX)

**S3 File. Moderation for medication use and help-seeking.**
(DOCX)

**S4 File. App user ratings.**
(DOCX)

**S5 File. CONSORT checklist.**
(DOCX)

**S6 File. Participant characteristics (Engager groups).**
(DOCX)

## Author contributions

**Conceptualization:** Mark Deady.

**Data curation:** Mikayla Gregory, Quincy J. J. Wong, Denise Meuldijk.

**Formal analysis:** Quincy J. J. Wong.

**Funding acquisition:** Mark Deady, Richard Bryant, Samuel B. Harvey.

**Investigation:** Mark Deady, Denise Meuldijk, Daniel A. J. Collins.

**Methodology:** Mark Deady, Mikayla Gregory, Denise Meuldijk, Daniel A. J. Collins.

**Project administration:** Mark Deady.

**Supervision:** Mark Deady, Richard Bryant, Samuel B. Harvey.

**Writing – original draft:** Mark Deady, Mikayla Gregory, Quincy J. J. Wong, Daniel A. J. Collins.

**Writing – review & editing:** Mark Deady, Quincy J. J. Wong, Denise Meuldijk, Daniel A. J. Collins, Lasse B. Sander, Richard Bryant, Samuel B. Harvey.

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
