## [Decision Letter · Decision Letter 0]

18 Jun 2025

Dear Dr. Deady,

Thank you for submitting your manuscript to PLOS ONE. After careful consideration, we feel that it has merit but does not fully meet PLOS ONE’s publication criteria as it currently stands. Therefore, we invite you to submit a revised version of the manuscript that addresses the points raised during the review process.

We look forward to receiving your revised manuscript.

Kind regards,

Fatma Refaat Ahmed, Ph.D.

Academic Editor

PLOS ONE

Journal Requirements:

2.  We note that the original protocol that you have uploaded as a Supporting Information file contains an institutional logo. As this logo is likely copyrighted, we ask that you please remove it from this file and upload an updated version upon resubmission.

“MD, DM, DAJC, SBH developed the Build Back Better app, they recieve no benefit from this program.”

4. In the online submission form, you indicated that “Data cannot be shared publicly as this was not clearly stated within the orginal ethics. However, deidentified data can be made available upon reasonable request to the authors.”

Reviewers' comments:

Reviewer's Responses to Questions

**Comments to the Author**

1. Is the manuscript technically sound, and do the data support the conclusions?

Reviewer #1: No

Reviewer #2: Yes

Reviewer #3: Partly

Reviewer #4: Yes

2. Has the statistical analysis been performed appropriately and rigorously?

Reviewer #1: Yes

Reviewer #2: Yes

Reviewer #3: No

Reviewer #4: No

3. Have the authors made all data underlying the findings in their manuscript fully available?

Reviewer #1: No

Reviewer #2: Yes

Reviewer #3: No

Reviewer #4: No

4. Is the manuscript presented in an intelligible fashion and written in standard English?

Reviewer #1: Yes

Reviewer #2: Yes

Reviewer #3: Yes

Reviewer #4: Yes

Reviewer #1: Thanks for the opportunity to review this manuscript. Overall, the science is fine and its well written.

I have significant concerns about the rationale for the study, and the interpretation of the findings, as follows:

The introduction, and therefore the discussion, fails to address the well know engagement problem in digital mental health. There are hundreds of thousands of apps, less than 5% are used at 30 days, there are well-known severe issues with engagement of digital mental health apps (SEE- https://pubmed.ncbi.nlm.nih.gov/29871870/). The rationale for this study must be made in light of this problem. What did this study do to address the known challenges of engagement, and where it did not address these issues, why?

The manuscript conflates the method of past studies and this study. Was co-design done as part of this project? If so, please elaborate on how, when, what, etc in the methods and results. If not, please move the co-design findings out of the method and into the introduction, clearly expand on how the co-design was conducted, and critically, why? Headgear was an effective intervention, you co-designed a new intervention (why? how?) that was no longer effective. This must be explained in the introduction and revisited in the discussion. The main point of the discussion that is missing is how co-design improvements to Headgear resulted in a non-effective intervention. Please interrogate the co-design approach, the changes that were made, and what the broader digital mental health literature can tell us to explain this unexpected result.

The power calculation describes a prevention study, the title and method describes a treatment study. In addition, why wasnt the Headgear study used to power this study?

The study cycles between the terms usability, satisfaction, usage, acceptability with inconsistency. (SEE - https://pubmed.ncbi.nlm.nih.gov/30914003/) The aims in the introduction need to be clearer about how these constructs will be assessed and what thresholds for satisfaction and usability will be set. It is not sufficient to say satisfaction was higher, what was the pre-study threshold? The method describes qualitative data that is not presented, and all of the 12 user

engagement and app-related feedback questions results are not presented. Table 6 presents means that are difficult to interpret- what does a few hundred opens actually mean?

Reviewer #2: Thank you for the opportunity to review the manuscript entitled "Efficacy of a smartphone app to improve mental health among emergency service workers: A randomised controlled trial". It is well-written and addresses an important topic. Below are some comments for consideration:

Lines 79-84: Few studies have been conducted on electronic support workers (ESW) using digital interventions. What might be the reasons for this? Is it due to the demanding nature of their work, which may limit the use of electronic devices during work hours? This is a crucial point, as it could impact adherence to the intervention. More evidence is needed to explore the behavioral habits regarding technology use as a tool for addressing mental health issues in this group.

Line 88: Please provide a brief overview of the acceptability, acceptance, and preferences of ESW regarding the pilot study outcomes. What changes were implemented from the pilot results to inform the randomized controlled trial (RCT)?

Lines 130-132: All study participants were instructed to download the Build Back Better smartphone app, including a login code tailored to their group allocation. Can participants log in on multiple devices, as long as they enter the correct code? If so, this raises concerns about the potential for participants to access the app on multiple devices or on devices belonging to others, which could lead to contamination.

Line 159: Regarding the content of the intervention—1) mindfulness, 2) healthy coping, 3) managing thoughts, and 4) valued actions—why were these aspects selected? How was this content validated? Please describe the intervention in detail, including session structure, duration, and how participants can seek support or ask questions related to this content.

Lines 166-167: The app provides links and phone numbers to various mental health and workplace support services. What is the purpose of including this content? How can you determine whether the effectiveness observed in participants is attributable to the app intervention or if it results from direct contact with mental health services?

Line 177: What are the psychometric properties of the instruments used in the study?

Table 6: In the app activity usage patterns, the monthly open frequency (average number of times an activity is accessed) is reported. How much time do participants spend on the app? What measures are in place to ensure participants engage with all the content within the app?

Reviewer #3: The present paper reports on a two-arm repeated measures randomized trial aimed at improving mental health among emergency service workers. The intervention arm includes the full version of the Build Back Better app, while the control arm involves symptom tracking only. Each participant is assessed at three-time points: baseline, 4 weeks post-baseline, and 3-month follow-up.

The manuscript has several methodological concerns that warrant attention:

Sample Size Calculation in Repeated Measures Design:

Sample size estimation for repeated measures trials is complex and typically requires specifying the intra-class correlation (ICC) or within-subject correlation. This critical parameter appears to be missing in the current manuscript. Without it, the sample size justification is incomplete.

Multiple Primary Outcomes:

Pages 8–10 describe six different outcomes, all of which seem to be treated as primary. This raises serious concerns about inflated Type I error rates due to multiple comparisons. Unless corrected (e.g., through statistical adjustment or designating a single primary outcome), the study is likely underpowered to detect significant effects across all six measures. The absence of a clearly defined primary outcome is a major design flaw.

High Dropout Rate and Underpowering:

The reported dropout rate of 30% is substantial. While the authors have attempted to adjust the sample size accordingly, Figure 1 (CONSORT diagram) shows only a total of 171 participants completing the trial. If this is the number used for analysis at the primary endpoint, the study may be underpowered even for a single outcome. Additionally, Figure 1 is difficult to read and should be revised for clarity.

Handling of Dropout and Partial Data:

It is unclear how data from participants who dropped out but provided partial data were handled. Were these data included in any form of mixed-effects model or imputed? Clarification is needed on how missing data were addressed, particularly for a longitudinal design.

Power Analysis Inconsistency:

On page 16, the authors report testing an interaction term for statistical significance, yet the power analysis on page 12 appears to be based on detecting a main effect only. A valid power analysis must align with the actual statistical model used in the analysis. If the interaction term is of interest, it should have been included in the original power estimation.

Conclusion:

I believe the paper has the potential to make a valuable contribution, but in its current form, it requires major revision. A well-defined primary outcome and a properly aligned power analysis would go a long way in strengthening the manuscript. Additionally, clarifying how missing data were handled and revising the presentation of the CONSORT diagram would enhance transparency and rigor.

Reviewer #4: This paper reports a primary analysis of an RCT comparing a digital intervention with a monitoring only digital intervention for emergency service workers. There was limited evidence for differential efficacy between the two groups, with both showing comparable positive outcomes across a range of primary and secondary outcomes. The paper is well written, however, in my opinion, deeper reflection on the drop out and engagement patterns, and what this means for digital interventions in this space (e.g., early intervention/indicated prevention in ESWs).

Major Comments

- Method: Please explain the rationale for choosing a self-initiated mood/activity log rather than a protocol typically used in experience sampling studies (i.e., questionnaires sent at fixed/random/semi-random times each day?). Whilst event-related approaches are common for behaviours like alcohol use, they are less common for internal experiences like mood, and studies usually have low compliance in these designs. This could also be a Discussion point for low engagement in both conditions.

- Data analysis page 13: The analysis plan includes multiple testing. What is the rationale for not correcting for multiple tests? Does controlling for multiple testing influence the pattern of results?

- Data analysis page 13 line 289-291. Explain the thresholds for no, minimal and engagers. They seem quite arbitrary (e.g., why not use a continuous variable?) and it would be worth including this as a broader point in the discussion (incl the lack of guidelines around what acceptable engagement actually is in digital interventions and monitoring (e.g., see experience sampling literature)

- Discussion: The authors acknowledge the high amount of drop out and from Baseline to T1 and T2, and low engagement, in both conditions (but particularly for the full intervention condition) in the Discussion. Related to the point above, perhaps the instruction given to participants influenced engagement (Page 6 line 132: Pts were asked to use the app consistently for 30 days”). On the one hand, this instruction is vague. On the other hand, if pts were instructed to use the app/s flexibly, perhaps the low compliance reflects just that (e.g., they used it when they needed to). The Discussion can be enhanced by elaborating on these points in the context of the specific intervention/s but also more broadly in terms of what it means for “preventive” digital interventions (or interventions more generally) for this population.

- Discussion: The disconnect between very low engagement and relatively positive app-feedback is a surprising result which has not been adequately addressed. I recommend adding a discussion point about why this disconnect might be present in this group and whether positive feedback is enough to recommend broader use/scale-up (i.e., what can we really do with this information given that the full intervention was not really used and did not improve most outcomes compared to the control?). Relatedly, is this disconnect common in digital intervention studies, or within this sample?

Minor Comments

- The authors mention in the introduction and methods that the app was co-designed with ESWs. Please specify the level of co-design according to established models – e.g., did they actively collaborate with and make decisions together with the research team or was still a top down approach to development (with researchers/clinicians making the final decisions). This could add to the authors point in the Discussion about app development and dropout/engagement.

- Page 4 line 78-79: “However, less is known on the equivalence of app-based interventions compared to 79 browser-based digital programs [21].” This makes it seem like the trial will be comparing these 2 variants or focusing on browser-based programs. I recommend removing or revising so that it aligns with the approach of the current study.

- Method: Please include examples of the mood and activity tracking questions (e.g., in supp materials)

- Methods: Usability and acceptability are broad terms that are defined differently across studies and within implementation literature. Please define what these concepts and link to previous literature. Relatedly, please describe how the engagement and app-related feedback questionnaire was developed (was it based on a previous study? Was it developed as a bespoke questionnaire?)

- Results: Given that the purpose of the RCT is to compare the effect of the two groups across time, rather than within-group changes, I suggest summarising these results clearly first.

- Page 16 line 341. In the primary outcomes section, it is written that the interaction “seldom” reached significance. No statistical tests for the interaction were significant, please reword to remove ambiguity.

- Discussion page 29. Please revise this sentence so that that the meaning is clear: “Nevertheless, although both mindfulness and behavioural activation formed part of the Build Back Better content, as part of the series of skills presented and may not have been engaged with in sufficient depth to elicit effect.”

**Do you want your identity to be public for this peer review?** For information about this choice, including consent withdrawal, please see our Privacy Policy

Reviewer #1: No

Reviewer #2: **Yes:** Cho Lee Wong

Reviewer #3: No

Reviewer #4: No

---

## [Author Response · Author response to Decision Letter 1]

24 Sep 2025

Fatma Refaat Ahmed, Ph.D.

Academic Editor

PLOS ONE

Dear Assoc Professor Refaat Ahmed,

Thank you for inviting us to resubmit this revised manuscript to PlosOne. In the attached manuscript we have addressed the items raised by the editorial team and peer reviewers and we have included our responses to the reviewer comments below.

Kind Regards,

Mark Deady, PhD

Editor comments:

Journal Requirements:

The manuscript has been updated to meet PLOS ONE's style requirements as follows:

• Changes to formatting of headings throughout.

• Removal of “Key words” and “Funding” on p. 3.

• Capitalisation changed to lower case where relevant.

• Continuation of line numbers throughout.

• Figures listed as “Fig 1”, “Fig 2”, etc.

• Tables included directly after the paragraph in which they are first cited.

File names have also been updated to meet the journal’s requirements.

2. We note that the original protocol that you have uploaded as a Supporting Information file contains an institutional logo. As this logo is likely copyrighted, we ask that you please remove it from this file and upload an updated version upon resubmission.

The institutional logo has been removed from the trial protocol and an updated version has been uploaded for inclusion as a Supporting Information file (“S3_File”).

“MD, DM, DAJC, SBH developed the Build Back Better app, they recieve no benefit from this program.”

Thank you kindly for this information. Please update our Competing Interests statement as follows:

“MD, DM, DAJC, and SBH developed the Build Back Better app. They receive no financial benefit from this program. This does not alter our adherence to PLOS ONE policies on sharing data and materials.”

4. In the online submission form, you indicated that “Data cannot be shared publicly as this was not clearly stated within the orginal ethics. However, deidentified data can be made available upon reasonable request to the authors.”

Thank you for drawing our attention to this. We would like to request exemption from making the data publicly available for ethical reasons, as detailed in our updated Data Availability statement:

“The data underlying the findings described in this manuscript are not publicly available due the human research ethics committee conditions of approval for the study. Other researchers who wish to access the data must provide written evidence of their own human research ethics approval before the data can be shared for secondary research purposes. Deidentified data can be made available upon reasonable request to the corresponding author (MD) along with evidence of human research ethics approval.”

Captions for Supporting Information files have now been included on the final page of the manuscript. The in-text citation referring to “supplementary materials” has also been updated to match the updated file name “S1_File”.

Reviewers' comments:

Reviewer #1:

Thanks for the opportunity to review this manuscript. Overall, the science is fine and its well written.

I have significant concerns about the rationale for the study, and the interpretation of the findings, as follows:

1. The introduction, and therefore the discussion, fails to address the well know engagement problem in digital mental health. There are hundreds of thousands of apps, less than 5% are used at 30 days, there are well-known severe issues with engagement of digital mental health apps (SEE- https://pubmed.ncbi.nlm.nih.gov/29871870/). The rationale for this study must be made in light of this problem. What did this study do to address the known challenges of engagement, and where it did not address these issues, why?

We agree with the issues raised by Reviewer 1. Indeed, this was the major rationale for the codesign process described below to tailor an existing evidence-based app to this new population. We have included greater detail both on this process and the rationale for it and the existing issues within the literature in this regard. We took a number of steps, informed by the literature, to enhance engagement as outlined below. As the Reviewer highlights Torous and colleagues note several strategies to improve engagement including the involvement of end users in the conception, design and testing of apps (which we have provided further detail on within the manuscript). Furthermore, as this paper states “apps are not the only sphere where engagement can be difficult. Adherence and engagement with information, monitoring, medications and psychotherapy are challenging problems encountered with both the face-to-face or via a digital interface.” Unguided apps/digital interventions do, of course, allow for both the freedom, and with it, the obstacle of ease of disengagement, which can occur without breaking any social contract of accountability. In our case ultimately, engagement remained a significant problem despite considered steps to develop an engaging solution that met user needs, highlighting the ongoing challenges in this space. This finding has recently been demonstrated in a metaanalysis [1], highlighting the considerable challenges inherent in optimising engagement with digital interventions.

“In order to adapt the program to the needs of ESWs an iterative codesign process occurred with ESWs, mental health experts including psychiatrists and clinical psychologists, consulted alongside design, user experience, and IT teams. Workshops and multiple rounds of user testing and revision took place during this phase with 12 ESWs to tailor this product. This codesign was deemed essential to meet the unique needs of the population. In order to overcome known challenges of engagement ([2]), several specific techniques associated with increased engagement were employed. These included personalised content where users selected area of need, data visualisation (tracking), reminders/push notifications, educational information, self-monitoring, and goal-setting features [3].”

2. The manuscript conflates the method of past studies and this study. Was co-design done as part of this project? If so, please elaborate on how, when, what, etc in the methods and results. If not, please move the co-design findings out of the method and into the introduction, clearly expand on how the co-design was conducted, and critically, why? Headgear was an effective intervention, you co-designed a new intervention (why? how?) that was no longer effective. This must be explained in the introduction and revisited in the discussion. The main point of the discussion that is missing is how co-design improvements to Headgear resulted in a non-effective intervention. Please interrogate the co-design approach, the changes that were made, and what the broader digital mental health literature can tell us to explain this unexpected result.

We have modified the methods and introduction to better reflect the codesign occurring outside of this specific RCT study. Co-design was an iterative process working with ESWs, mental health professionals, design, user experience, and IT teams.

Although the reviewer is correct that HeadGear being effective was important in the genesis for the project, simply reusing this app in such a unique population was viewed as inadequate [4, 5]. It was believed that this process would lead to enhanced effectiveness and improved outcomes by addressing individual needs and preferences. At least in part due to better engagement and increased motivation, and salience of message. To the reviewer’s query 1, engagement was a major consideration and something we hoped to improve. As the reviewer points out, in our case this did not occur (although direct comparison is missing). We have further elaborated on this in the discussion. However, there are other trial differences that may explain the different findings.

“…Finally, in line with recommendations for codesign of ESW interventions [6], representatives from this population were involved in ongoing consultation throughout the app development process, and on the basis of iterative codesign Build Back Better moved from a sequential 30-day once-daily format to an unstructured, self-directed format which may have impacted the effectiveness of the app [7]. It is unclear exactly what drove this null finding especially where the clinical basis for the app was previously associated with effect [8]. However, two major distinctions were the baseline clinical profile of users and the freeform app format. Specifically, this format change may have led to poorer direction within the app. Although users in all these studies were directed to use the app consistently for 30 days”, this instruction is likely to be considerably clearer where the app structure follows this instruction. Instead, it is more likely present study participants (especially as they were symptomatic at baseline) may have used the app flexibility (e.g., when they needed to), which is unlikely to bring about sustained effect.”

3. The power calculation describes a prevention study, the title and method describes a treatment study. In addition, why wasnt the Headgear study used to power this study?

The sample was different in terms of risk and outcome of interest. Where the Headgear study was universal in nature looking at depressive symptom change with baseline restrictions around symptom severity, the current trial was aimed to improve psychological distress in a group experiencing at least moderate levels of psychological distress. However, we an error that we have since corrected in the section detailing this calculation. Nevertheless, the study was under powered upon completion due to insufficient recruitment, we have raised this as a point of discussion.

4. The study cycles between the terms usability, satisfaction, usage, acceptability with inconsistency. (SEE - https://pubmed.ncbi.nlm.nih.gov/30914003/) The aims in the introduction need to be clearer about how these constructs will be assessed and what thresholds for satisfaction and usability will be set. It is not sufficient to say satisfaction was higher, what was the pre-study threshold? The method describes qualitative data that is not presented, and all of the 12 user engagement and app-related feedback questions results are not presented. Table 6 presents means that are difficult to interpret- what does a few hundred opens actually mean?

We have sought to clarify this throughout for consistency. We have also resolved inconsistency around language throughout. Our measure of usability was based off the Mobile Application Rating Scale. Although not exhaustive, these items within this validated app quality rating tool provide some insight into user experience of the app.

“Items were adapted from the Mobile Application Rating Scale [9], including ease of use, understanding of content, engagement and interest in the design and content, likelihood of recommending to others, and overall rating of the app and website. Further questions measured the subjective perception of improvement in mental fitness, and reasons for stopping app use. Participants also provided general feedback and suggestions via open-response questions. These measures have been used previously [10].”

We have further removed specific reference to this within the aims as the exploration of use patterns and feedback merely supports the main analysis, and the results do not sufficiently answer a unique question. We do not state that “satisfaction is higher” but we do feel it valuable to compare the groups on these feedback scores to determine differences in experience of the two versions of the app. However, we note that there are no established thresholds in this space to reliably determine whether the app was viewed as “adequately” helpful.

In light of Reviewer 1’s comments we have removed Table 6 from the manuscript and in place have provided supplemental material detailing full list of feedback items used. We have also modified the app use and feedback section to provide greater detail of the findings in regard to this feedback.

“Around half of respondents (129/254; 50.79%) reported stopping use of the app before completing the trial. A higher proportion of the full intervention group (61/103; 59.22%) ceased app use compared to the tracking-only group (68/151; 45.03%). The most common reason for stopping in the full intervention group was lack of time (23/61; 37.70%), whereas the most common reason in the tracking-only group was loss of interest (29/68; 42.65%),

When asked to nominate their favourite app feature, full intervention respondents most frequently noted ease of use/navigation, the wide variety of available content, and the daily tracker. Of the four key content areas, mindfulness was favoured the most.

In terms of least favourite app feature, full intervention respondents most frequently mentioned lack of reminders to use the app (this function was available but required setting changes to turn on), and lack of engaging or relevant content. A small minority reported technical issues (such as inability to access offline) and difficulty with navigation/ease of use.”

Reviewer #2:

Thank you for the opportunity to review the manuscript entitled "Efficacy of a smartphone app to improve mental health among emergency service workers: A randomised controlled trial". It is well-written and addresses an important topic. Below are some comments for consideration:

1. Lines 79-84: Few studies have been conducted on electronic support workers (ESW) using digital interventions. What might be the reasons for this? Is it due to the demanding nature of their work, which may limit the use of electronic devices during work hours? This is a crucial point, as it could impact adherence to the intervention. More evidence is needed to explore the behavioral habits regarding technology use as a tool for addressing mental health issues in this group.

We have sought to further explore this in the revised manuscript. Primarily, this is the result of limited interventions specifically designed for this population more broadly. Certainly, there are limitations related to on-shift use but these also apply to most frontline workforces (e.g. health

---

## [Decision Letter · Decision Letter 1]

5 Nov 2025

Dear Dr. Deady,

We look forward to receiving your revised manuscript.

Kind regards,

Fatma Refaat Ahmed, Ph.D.

Academic Editor

PLOS ONE

Journal Requirements:

Reviewers' comments:

Reviewer's Responses to Questions

**Comments to the Author**

Reviewer #5: All comments have been addressed

Reviewer #6: (No Response)

2. Is the manuscript technically sound, and do the data support the conclusions?

Reviewer #5: Yes

Reviewer #6: Partly

3. Has the statistical analysis been performed appropriately and rigorously?

Reviewer #5: Yes

Reviewer #6: N/A

4. Have the authors made all data underlying the findings in their manuscript fully available?

Reviewer #5: No

Reviewer #6: Yes

5. Is the manuscript presented in an intelligible fashion and written in standard English?

Reviewer #5: Yes

Reviewer #6: Yes

Reviewer #5: Thank you for the opportunity to conduct a second-look review. I was asked to comment on a manuscript that has already undergone extensive peer review, and I have therefore focused my assessment on whether the authors have addressed the editor’s and reviewers’ first-round comments. It would be unfair of me to provide fresh round of comments at this stage. The authors have, in my view, largely addressed the prior comments through concrete textual revisions, clearer reporting, and additional analyses. A small number of issues are acknowledged rather than fully resolved.

Specific points addressed

* formatting, file naming, and figure/table placement have been corrected.

* the statement has been clarified and the authors request an ethics-based exemption; I note this requires an editorial decision as it int clear why the committee are blocking.

* the description of the power analysis has been aligned with the tests actually conducted; explicit between-group results at follow-ups have been added.

* handling is now described (linear mixed models with maximum likelihood under a MAR assumption).

* the results and discussion have been adjusted to acknowledge low engagement and minimal between-group differences, with implications for unguided digital interventions made more explicit.

On balance, the first-round comments have been adequately addressed. I recommend acceptance subject to the editorial team being satisfied with the data availability wording and any minor textual edits they may request.

Reviewer #6: The authors have made a commendable effort to address the concerns raised in the previous review round and the manuscript has been improved. However, some questions remain.

- Please clarify which software or method was used for the power analysis (e.g., G*Power, R package, or other).

- The use of linear mixed models (LMMs) with maximum likelihood estimation is methodologically appropriate for repeated measures data; however, the authors indicate that models “did not adjust for baseline scores.” The rationale for this decision should be provided.

- Little’s MCAR test was non-significant (p = 1.00), suggesting that there was no systematic missingness. However, attrition rates differed significantly between conditions at both follow-ups. This discrepancy indicates that data were unlikely to be missing completely at random (MCAR).

- Including descriptive comparisons of baseline characteristics across engagement groups (non-engagers, minimal engagers, and engagers) would enhance the interpretability of the findings and provide important context regarding potential differences in initial symptom severity or demographics.

- The discussion section is well written, appropriately acknowledging the null results and relevant contextual factors affecting the target population. In fact, the authors noted that the active control condition (mood and behavior tracking) is an important feature of this trial and likely contributed to improvements in both arms. The implications of using an active comparator deserve greater discussion.

- Exploratory findings, particularly the benefits for PTSD symptoms and help-seeking participants, should be presented more cautiously, given the small subsample and missing data.

- Finally, the conclusion that unguided digital interventions may not suit this population is reasonable but could be complemented by more constructive recommendations for future development (hybrid or guided models? adaptive tailoring based on symptom severity?).

**Do you want your identity to be public for this peer review?** For information about this choice, including consent withdrawal, please see our Privacy Policy

Reviewer #5: **Yes:** Dr Daniel Leightley

Reviewer #6: No

---

## [Author Response · Author response to Decision Letter 2]

10 Nov 2025

Fatma Refaat Ahmed, Ph.D.

Academic Editor

PLOS ONE

Dear Assoc Professor Refaat Ahmed,

Thank you for inviting us to resubmit this revised manuscript to PlosOne. In the attached manuscript we have addressed the items raised by the editorial team and peer reviewers and we have included our responses to the reviewer comments below.

Kind Regards,

Mark Deady, PhD

Comments to the Author

1. If the authors have adequately addressed your comments raised in a previous round of review and you feel that this manuscript is now acceptable for publication, you may indicate that here to bypass the “Comments to the Author” section, enter your conflict of interest statement in the “Confidential to Editor” section, and submit your "Accept" recommendation.

Reviewer #5: All comments have been addressed

Reviewer #6: (No Response)

NA

2. Is the manuscript technically sound, and do the data support the conclusions?

Reviewer #5: Yes

Reviewer #6: Partly

NA

3. Has the statistical analysis been performed appropriately and rigorously?

Reviewer #5: Yes

Reviewer #6: N/A

NA

4. Have the authors made all data underlying the findings in their manuscript fully available?

Reviewer #5: No

Reviewer #6: Yes

We have now provided this in repository: https://doi.org/10.26190/unsworks/31622

Which will be live once the manuscript is published.

5. Is the manuscript presented in an intelligible fashion and written in standard English?

Reviewer #5: Yes

Reviewer #6: Yes

NA

6. Review Comments to the Author

Reviewer #5: Thank you for the opportunity to conduct a second-look review. I was asked to comment on a manuscript that has already undergone extensive peer review, and I have therefore focused my assessment on whether the authors have addressed the editor’s and reviewers’ first-round comments. It would be unfair of me to provide fresh round of comments at this stage. The authors have, in my view, largely addressed the prior comments through concrete textual revisions, clearer reporting, and additional analyses. A small number of issues are acknowledged rather than fully resolved.

Specific points addressed

* formatting, file naming, and figure/table placement have been corrected.

* the statement has been clarified and the authors request an ethics-based exemption; I note this requires an editorial decision as it int clear why the committee are blocking.

* the description of the power analysis has been aligned with the tests actually conducted; explicit between-group results at follow-ups have been added.

* handling is now described (linear mixed models with maximum likelihood under a MAR assumption).

* the results and discussion have been adjusted to acknowledge low engagement and minimal between-group differences, with implications for unguided digital interventions made more explicit.

On balance, the first-round comments have been adequately addressed. I recommend acceptance subject to the editorial team being satisfied with the data availability wording and any minor textual edits they may request.

NA

Reviewer #6: The authors have made a commendable effort to address the concerns raised in the previous review round and the manuscript has been improved. However, some questions remain.

- Please clarify which software or method was used for the power analysis (e.g., G*Power, R package, or other).

We have now included this.

G*Power was used to calculate power.

- The use of linear mixed models (LMMs) with maximum likelihood estimation is methodologically appropriate for repeated measures data; however, the authors indicate that models “did not adjust for baseline scores.” The rationale for this decision should be provided.

The reviewer raises a good point. Randomisation in this trial resulted in comparable groups of participants in trial arms at baseline (see Table 2). As such, our analytical models did not adjust for baseline scores. We have now revised the Data Analysis section to include this point:

Models did not adjust for baseline scores as randomisation in this trial resulted in comparable groups of participants in trial arms at baseline (see Table 2).

- Little’s MCAR test was non-significant (p = 1.00), suggesting that there was no systematic missingness. However, attrition rates differed significantly between conditions at both follow-ups. This discrepancy indicates that data were unlikely to be missing completely at random (MCAR).

This is a good point. We have now added further information in the revised paper to provide greater clarification in the Results section:

Across all outcome measures, Little’s Missing Completely at Random (MCAR) test was not significant, χ2(3564) = 2379.28, p = 1.00, indicating missing data were MCAR. However, attrition from assessment was greater in the full intervention condition than the tracking-only condition at 1-month follow-up (72.3% vs 58.9%; χ2(1) = 17.52, p < .001) and at 3-month follow-up (83.6% vs 77.5%; χ2(1) = 5.29, p = .027). In addition, missingness was significantly related to variables in the dataset (e.g., GAD-7 item and AUDIT-C item scores; ps < .046). Taken together, these results suggest missing data were more plausibly missing at random (MAR), and this aligns with the missing-at-random assumption of the linear mixed models approach for analyses.

- Including descriptive comparisons of baseline characteristics across engagement groups (non-engagers, minimal engagers, and engagers) would enhance the interpretability of the findings and provide important context regarding potential differences in initial symptom severity or demographics.

We have now included this descriptive data as an appendix table (S7).

- The discussion section is well written, appropriately acknowledging the null results and relevant contextual factors affecting the target population. In fact, the authors noted that the active control condition (mood and behavior tracking) is an important feature of this trial and likely contributed to improvements in both arms. The implications of using an active comparator deserve greater discussion.

We have attempted to further discuss this within the manuscript, whilst being sensitive to word count restrictions. We also considered the overall effects across the two groups in determining the impact of the active control.

Indeed, there is evidence to suggest mood monitoring is associated with symptom improvement particularly in the case of depression [1, 2], with active controls of this nature associated with significantly smaller effect sizes compared to other forms of control [3]. This tool may have been further enhanced by the inclusion of other forms of monitoring (e.g., activity, sleep) which may have allowed users to recognise patterns in these factors and improve self-regulation. This adds to the ongoing debate around accuracy of evidence for psychological intervention in the context of appropriate control conditions, and in fact what such conditions are [4]. Further comparisons of monitoring and specific self-guided program elements are required to better understand the utility of this form of control condition and what therapeutic impacts it may have.

- Exploratory findings, particularly the benefits for PTSD symptoms and help-seeking participants, should be presented more cautiously, given the small subsample and missing data.

We have now specifically noted the methodological limitations of these findings.

However, the small sample size and missing data mean these findings should be interpreted with caution.

- Finally, the conclusion that unguided digital interventions may not suit this population is reasonable but could be complemented by more constructive recommendations for future development (hybrid or guided models? adaptive tailoring based on symptom severity?).

We have now added further context to this conclusion

These alternatives might include blended care or more guided modes of delivery (e.g., [5]). The use of adaptive and individually tailored approaches may also improve engagement.

We thank the reviewers for their insightful and perspicacious review and suggestions.

Regards,

Mark Deady and authorship team

1. van der Watt ASJ, Odendaal W, Louw K, Seedat S. Distant mood monitoring for depressive and bipolar disorders: a systematic review. BMC Psychiatry. 2020;20(1):383. doi: 10.1186/s12888-020-02782-y.

2. Dubad M, Winsper C, Meyer C, Livanou M, Marwaha S. A systematic review of the psychometric properties, usability and clinical impacts of mobile mood-monitoring applications in young people. Psychological Medicine. 2018;48(2):208-28. Epub 2017/06/23. doi: 10.1017/S0033291717001659.

3. Linardon J, Cuijpers P, Carlbring P, Messer M, Fuller‐Tyszkiewicz M. The efficacy of app‐supported smartphone interventions for mental health problems: A meta‐analysis of randomized controlled trials. World Psychiatry. 2019;18(3):325-36.

4. Cristea IA. The waiting list is an inadequate benchmark for estimating the effectiveness of psychotherapy for depression. Epidemiol Psychiatr Sci. 2019;28(3):278-9. Epub 2018/11/28. doi: 10.1017/s2045796018000665. PubMed PMID: 30479243; PubMed Central PMCID: PMCPMC6998910.

5. Deady M, Collins DAJ, Azevedo S, Stech E, Harrison A, Broomfield C, et al. Integration of a smartphone app with posttraumatic stress disorder treatment for frontline workers: a pilot study. Australian Journal of Psychology. 2024;76(1):2399112. doi: 10.1080/00049530.2024.2399112.

---

## [Decision Letter · Decision Letter 2]

22 Jan 2026

Efficacy of a smartphone app to improve mental health among emergency service workers: A randomised controlled trial

PONE-D-25-23028R2

Dear Dr. Deady,

We’re pleased to inform you that your manuscript has been judged scientifically suitable for publication and will be formally accepted for publication once it meets all outstanding technical requirements.

Kind regards,

Fatma Refaat Ahmed, Ph.D.

Academic Editor

PLOS One

Additional Editor Comments (optional):

Reviewers' comments:

Reviewer's Responses to Questions

**Comments to the Author**

Reviewer #6: All comments have been addressed

Reviewer #7: All comments have been addressed

2. Is the manuscript technically sound, and do the data support the conclusions?

Reviewer #6: Yes

Reviewer #7: Yes

3. Has the statistical analysis been performed appropriately and rigorously?

Reviewer #6: Yes

Reviewer #7: Yes

4. Have the authors made all data underlying the findings in their manuscript fully available?

Reviewer #6: Yes

Reviewer #7: Yes

5. Is the manuscript presented in an intelligible fashion and written in standard English?

Reviewer #6: Yes

Reviewer #7: Yes

Reviewer #6: The authors have addressed all the comments and concerns raised in the prior review round. Clarifications regarding the power analysis, baseline adjustment rationale, and handling of missing data have been included. They also have added descriptive comparisons of engagement groups as requested which improves the contextualization of results. The manuscript is clearly written, technically sound, and the conclusions are appropriately matched to the data. Moreover, they also have made available the dataset of the study.

I have no further concerns, and I consider the manuscript suitable for publication.

Reviewer #7: I have been invited to conduct a review on this paper, which has already undergone two previous rounds of review with other reviewers. Therefore, I am focusing my comments and ratings on whether the authors have addressed previous comments or concerns from the previous reviewers. In my opinion, the authors have adequately addressed the previous comments, with a minimal data set now being made publicly available. I recommend acceptance.

**Do you want your identity to be public for this peer review?** For information about this choice, including consent withdrawal, please see our Privacy Policy

Reviewer #6: No

Reviewer #7: No

---

## [Editor Report · Acceptance letter]

PONE-D-25-23028R2

PLOS One

Dear Dr. Deady,

I'm pleased to inform you that your manuscript has been deemed suitable for publication in PLOS One. Congratulations! Your manuscript is now being handed over to our production team.

Kind regards,

on behalf of

Dr. Fatma Refaat Ahmed

Academic Editor

PLOS One